# Kullback-Leibler Proximal Variational Inference

**Mohammad Emtiyaz Khan**\*
Ecole Polytechnique Fédérale de Lausanne
Lausanne, Switzerland
emtiyaz@gmail.com

**Pierre Baqué**\*
Ecole Polytechnique Fédérale de Lausanne
Lausanne, Switzerland
pierre.baque@epfl.ch

**François Fleuret**
Idiap Research Institute
Martigny, Switzerland
francois.fleuret@idiap.ch

**Pascal Fua**
Ecole Polytechnique Fédérale de Lausanne
Lausanne, Switzerland
pascal.fua@epfl.ch

## Abstract

We propose a new variational inference method based on a proximal framework that uses the Kullback-Leibler (KL) divergence as the proximal term. We make two contributions towards exploiting the geometry and structure of the variational bound. First, we propose a KL proximal-point algorithm and show its equivalence to variational inference with natural gradients (e.g., stochastic variational inference). Second, we use the proximal framework to derive efficient variational algorithms for non-conjugate models. We propose a splitting procedure to separate non-conjugate terms from conjugate ones. We linearize the non-conjugate terms to obtain subproblems that admit a closed-form solution. Overall, our approach converts inference in a non-conjugate model to subproblems that involve inference in well-known conjugate models. We show that our method is applicable to a wide variety of models and can result in computationally efficient algorithms. Applications to real-world datasets show comparable performances to existing methods.

## 1   Introduction

Variational methods are a popular alternative to Markov chain Monte Carlo (MCMC) methods for Bayesian inference. They have been used extensively for their speed and ease of use. In particular, methods based on the evidence lower bound optimization (ELBO) are quite popular because they convert a difficult integration problem to an optimization problem. This reformulation enables the application of optimization techniques for large-scale Bayesian inference.

Recently, an approach called stochastic variational inference (SVI) has gained popularity for inference in conditionally-conjugate exponential family models [1]. SVI exploits the geometry of the posterior distribution by using natural gradients and uses a stochastic method to improve scalability. The resulting updates are simple and easy to implement.

Several generalizations of SVI have been proposed for general latent-variable models where the lower bound might be intractable [2, 3, 4]. These generalizations, although important, do not take the geometry of the posterior distribution into account.

In addition, none of these approaches exploit the structure of the lower bound. In practice, not all *factors* of the joint distribution introduce difficulty in the optimization. It is therefore desirable to treat "difficult" terms differently from "easy" terms.

In this context, we propose a splitting method for variational inference; this method exploits both the structure and the geometry of the lower bound. Our approach is based on the proximal-gradient framework. We make two important contributions. First, we propose a proximal-point algorithm that uses the Kullback-Leibler (KL) divergence as the proximal term. We show that the addition of this term incorporates the geometry of the posterior distribution. We establish the equivalence of our approach to variational methods that use natural gradients (e.g., [1, 5, 6]).

Second, following the proximal-gradient framework, we propose a splitting approach for variational inference. In this approach, we linearize difficult terms such that the resulting optimization problem is easy to solve. We apply this approach to variational inference on non-conjugate models. We show that linearizing non-conjugate terms leads to subproblems that have closed-form solutions. Our approach therefore converts inference in a non-conjugate model to subproblems that involve inference in well-known conjugate models, and for which efficient implementation exists.

## 2   Latent Variable Models and Evidence Lower-Bound Optimization

Consider a general latent-variable model with data vector $\mathbf{y}$ of length $N$ and the latent vector $\mathbf{z}$ of length $D$, following a joint distribution $p(\mathbf{y}, \mathbf{z})$ (we drop the parameters of the distribution from the notation). ELBO approximates the posterior $p(\mathbf{z}|\mathbf{y})$ by a distribution $q(\mathbf{z}|\boldsymbol{\lambda})$ that maximizes a lower bound to the marginal likelihood. Here, $\boldsymbol{\lambda}$ is the vector of parameters of the distribution $q$. As shown in (1), the lower bound is obtained by first multiplying and then dividing by $q(\mathbf{z}|\boldsymbol{\lambda})$, and then applying Jensen's inequality by using concavity of $\log$. The approximate posterior $q(\mathbf{z}|\boldsymbol{\lambda})$ is obtained by maximizing the lower bound with respect to $\boldsymbol{\lambda}$.

$$\log p(\mathbf{y}) = \log \int q(\mathbf{z}|\boldsymbol{\lambda}) \frac{p(\mathbf{y}, \mathbf{z})}{q(\mathbf{z}|\boldsymbol{\lambda})} \, d\mathbf{z} \geq \max_{\boldsymbol{\lambda}} \mathbb{E}_{q(\mathbf{z}|\boldsymbol{\lambda})} \left[ \log \frac{p(\mathbf{y}, \mathbf{z})}{q(\mathbf{z}|\boldsymbol{\lambda})} \right] := \mathcal{L}(\boldsymbol{\lambda}). \tag{1}$$

Unfortunately, the lower bound may not always be easy to optimize, e.g., some terms in the lower bound might be intractable or might admit a form that is not easy to optimize. In addition, the optimization can be slow when $N$ and $D$ are large.

## 3   The KL Proximal-Point Algorithm for Conjugate Models

In this section, we introduce a proximal-point method based on Kullback-Leibler (KL) proximal function and establish its relation to the existing approaches based on natural gradients [1, 5, 6]. In particular, for conditionally-conjugate exponential-family models, we show that each iteration of our proximal-point approach is equivalent to a step along the natural gradient.

The Kullback-Leibler (KL) divergence between two distributions $q(\mathbf{z}|\boldsymbol{\lambda})$ and $q(\mathbf{z}|\lambda')$ is defined as follows: $\mathbb{D}_{KL}[q(\mathbf{z}|\boldsymbol{\lambda}) \, \| \, q(\mathbf{z}|\boldsymbol{\lambda}')] := \mathbb{E}_{q(\mathbf{z}|\boldsymbol{\lambda})}[\log q(\mathbf{z}|\boldsymbol{\lambda}) - \log q(\mathbf{z}|\boldsymbol{\lambda}')]$. Using the KL divergence as the proximal term, we introduce a proximal-point algorithm that generates a sequence of $\boldsymbol{\lambda}_k$ by solving the following subproblems:

$$\text{KL Proximal-Point}: \ \boldsymbol{\lambda}_{k+1} = \arg\max_{\boldsymbol{\lambda}} \ \mathcal{L}(\boldsymbol{\lambda}) - \frac{1}{\beta_k} \mathbb{D}_{KL}[q(\mathbf{z}|\boldsymbol{\lambda}) \, \| \, q(\mathbf{z}|\boldsymbol{\lambda}_k)], \tag{2}$$

given an initial value $\boldsymbol{\lambda}_0$ and a bounded sequence of step-size $\beta_k > 0$,

One benefit of using the KL term is that it takes the geometry of the posterior distribution into account. This fact has lead to their extensive use in both the optimization and statistics literature, e.g., for speeding up the expectation-maximization algorithm [7, 8], for convex optimization [9], for message-passing in graphical models [10], and for approximate Bayesian inference [11, 12, 13].

**Relationship to the methods that use natural gradients:** An alternative approach to incorporate the geometry of the posterior distribution is to use natural gradients [6, 5, 1]. We now establish its relationship to our approach. The natural gradient can be interpreted as finding a descent direction that ensures a fixed amount of change in the distribution. For variational inference, this is equivalent to the following [1, 14]:

$$\arg\max_{\Delta\boldsymbol{\lambda}} \mathcal{L}(\boldsymbol{\lambda}_k + \Delta\boldsymbol{\lambda}), \ \text{ s.t. } \ \mathbb{D}_{KL}^{sym}[q(\mathbf{z}|\boldsymbol{\lambda}_k + \Delta\boldsymbol{\lambda}) \, \| \, q(\mathbf{z}|\boldsymbol{\lambda}_k)] \leq \epsilon, \tag{3}$$

where $\mathbb{D}_{KL}^{sym}$ is the symmetric KL divergence. It appears that the proximal-point subproblem (2) is related to a Lagrangian of the above optimization. In fact, as we show below, the two problems are equivalent for conditionally conjugate exponential-family models.

We consider the set-up described in [15], which is a bit more general than that of [1]. Consider a Bayesian network with nodes $\mathbf{z}_i$ and a joint distribution $\prod_i p(\mathbf{z}_i|\mathrm{pa}_i)$ where $\mathrm{pa}_i$ are the parents of $\mathbf{z}_i$. We assume that each factor is an exponential-family distribution defined as follows:

$$p(\mathbf{z}_i|\mathrm{pa}_i) := h_i(\mathbf{z}_i) \exp\left[\boldsymbol{\eta}_i^T(\mathrm{pa}_i)\mathbf{T}_i(\mathbf{z}_i) - A_i(\boldsymbol{\eta}_i)\right], \tag{4}$$

where $\boldsymbol{\eta}_i$ is the natural parameter, $\mathbf{T}_i(\mathbf{z}_i)$ is the sufficient statistics, $A_i(\boldsymbol{\eta}_i)$ is the partition function and $h_i(\mathbf{z}_i)$ is the base measure. We seek a factorized approximation shown in (5), where each $\mathbf{z}_i$ belongs to the same exponential-family distribution as the joint distribution. The parameters of this distribution are denoted by $\boldsymbol{\lambda}_i$ to differentiate them from the joint-distribution parameters $\boldsymbol{\eta}_i$. Also note that the subscript refers to the factor $i$, not to the iteration.

$$q(\mathbf{z}|\boldsymbol{\lambda}) = \prod_i q_i(\mathbf{z}_i|\boldsymbol{\lambda}_i), \text{ where } q_i(\mathbf{z}_i) := h_i(\mathbf{z}) \exp\left[\boldsymbol{\lambda}_i^T \mathbf{T}_i(\mathbf{z}_i) - A_i(\boldsymbol{\lambda}_i)\right]. \tag{5}$$

For this model, we show the following equivalence between a gradient-descent method based on natural gradients and our proximal-point approach. The proof is given in the supplementary material.

**Theorem 1.** *For the model shown in* (4) *and the posterior approximation shown in* (5)*, the sequence $\boldsymbol{\lambda}_k$ generated by the proximal-point algorithm of* (2) *is equal to the one obtained using gradient-descent along the natural gradient with step lengths $\beta_k/(1+\beta_k)$.*

**Proof of convergence :** Convergence of the proximal-point algorithm shown in (2) is proved in [8]. We give a summary of the results here. We assume $\beta_k = 1$, however the proof holds for any bounded sequence of $\beta_k$. Let the space of all $\boldsymbol{\lambda}$ be denoted by $\mathcal{S}$. Define the set $\mathcal{S}_0 := \{\boldsymbol{\lambda} \in \mathcal{S} : \underline{\mathcal{L}}(\boldsymbol{\lambda}) \geq \underline{\mathcal{L}}(\boldsymbol{\lambda}_0)\}$. Then, $\|\boldsymbol{\lambda}_{k+1} - \boldsymbol{\lambda}_k\| \to 0$ under the following conditions:

(A) Maximum of $\underline{\mathcal{L}}$ exist and the gradient of $\underline{\mathcal{L}}$ is continuous and defined in $\mathcal{S}_0$.

(B) The KL divergence and its gradient are continuous and defined in $\mathcal{S}_0 \times \mathcal{S}_0$.

(C) $\mathbb{D}_{KL}[q(\mathbf{z}|\boldsymbol{\lambda}) \,\|\, q(\mathbf{z}|\boldsymbol{\lambda}')] = 0$ only when $\boldsymbol{\lambda}' = \boldsymbol{\lambda}$.

In our case, the conditions (A) and (B) are either assumed or satisfied, and the condition (C) can be ensured by choosing an appropriate parameterization of $q$.

## 4 The KL Proximal-Gradient Algorithm for Non-conjugate Models

The proximal-point algorithm of (2) might be difficult to optimize for non-conjugate models, e.g., due to the non-conjugate factors. In this section, we present an algorithm based on the proximal-gradient framework where we first split the objective function into "difficult" and "easy" terms, and then, to simplify the optimization, linearize the difficult term. See [16] for a good review of proximal methods for machine learning.

We split the ratio $p(\mathbf{y}, \mathbf{z})/q(\mathbf{z}|\boldsymbol{\lambda}) \equiv c\,\tilde{p}_d(\mathbf{z}|\boldsymbol{\lambda})\tilde{p}_e(\mathbf{z}|\boldsymbol{\lambda})$, where $\tilde{p}_d$ contains all factors that make the optimization difficult, and $\tilde{p}_e$ contains the rest ($c$ is a constant). This results in the following split:

$$\underline{\mathcal{L}}(\boldsymbol{\lambda}) = \mathbb{E}_{q(\mathbf{z}|\boldsymbol{\lambda})}\left[\log\frac{p(\mathbf{y}, \mathbf{z}|\boldsymbol{\theta})}{q(\mathbf{z}|\boldsymbol{\lambda})}\right] := \underbrace{\mathbb{E}_{q(\mathbf{z}|\boldsymbol{\lambda})}[\log\tilde{p}_d(\mathbf{z}|\boldsymbol{\lambda})]}_{f(\boldsymbol{\lambda})} + \underbrace{\mathbb{E}_{q(\mathbf{z}|\boldsymbol{\lambda})}[\log\tilde{p}_e(\mathbf{z}|\boldsymbol{\lambda})]}_{h(\boldsymbol{\lambda})} + \log c, \tag{6}$$

Note that $\tilde{p}_d$ and $\tilde{p}_e$ can be un-normalized factors in the distribution. In the worst case, we can set $\tilde{p}_e(\mathbf{z}|\boldsymbol{\lambda}) \equiv 1$ and take the rest as $\tilde{p}_d(\mathbf{z}|\boldsymbol{\lambda})$. We give an example of the split in the next section.

The main idea is to linearize the difficult term $f$ such that the resulting problem admits a simple form. Specifically, we use a proximal-gradient algorithm that solves the following sequence of subproblems to maximize $\underline{\mathcal{L}}$ as shown below. Here, $\bigtriangledown f(\boldsymbol{\lambda}_k)$ is the gradient of $f$ at $\boldsymbol{\lambda}_k$.

$$\text{KL Proximal-Gradient: } \boldsymbol{\lambda}_{k+1} = \arg\max_{\boldsymbol{\lambda}} \boldsymbol{\lambda}^T \bigtriangledown f(\boldsymbol{\lambda}_k) + h(\boldsymbol{\lambda}) - \frac{1}{\beta_k}\mathbb{D}_{KL}[q(\mathbf{z}|\boldsymbol{\lambda}) \,\|\, q(\mathbf{z}|\boldsymbol{\lambda}_k)]. \tag{7}$$

Note that our linear approximation is equivalent to the one used in gradient descent. Also, the approximation is tight at $\boldsymbol{\lambda}_k$. Therefore, it does not introduce any error in the optimization, rather it only acts as a surrogate to take the next step. Existing variational methods have used approximations such as ours, e.g., see [17, 18, 19]. Most of these methods first approximate the $\log \tilde{p}_d(\mathbf{z}|\boldsymbol{\lambda})$ term by using a linear or quadratic approximation and then compute the expectation. As a result the approximation is not tight and can result in a bad performance [20]. In contrast, our approximation is applied directly to $\mathbb{E}[\log \tilde{p}_d(\mathbf{z}|\boldsymbol{\lambda})]$ and therefore is tight at $\boldsymbol{\lambda}_k$.

The convergence of our approach is covered under the results shown in [21]; they prove convergence of an algorithm more general algorithm than ours. Below, we summarize the results. As before, we assume that the maximum exists and $\mathcal{L}$ is continuous. We make three additional assumptions. First, the gradient of $f$ is $L$-Lipschitz continuous in $\mathcal{S}$, i.e., $||\bigtriangledown f(\boldsymbol{\lambda}) - \bigtriangledown f(\boldsymbol{\lambda}')|| \leq L||\boldsymbol{\lambda} - \boldsymbol{\lambda}'||, \forall \boldsymbol{\lambda}, \boldsymbol{\lambda}' \in \mathcal{S}$. Second, the function $h$ is concave. Third, there exists an $\alpha > 0$ such that,

$$(\boldsymbol{\lambda}_{k+1} - \boldsymbol{\lambda}_k)^T \bigtriangledown_1 \mathbb{D}_{KL}[q(\mathbf{z}|\boldsymbol{\lambda}_{k+1}) \| q(\mathbf{z}|\boldsymbol{\lambda}_k)] \geq \alpha \|\boldsymbol{\lambda}_{k+1} - \boldsymbol{\lambda}_k\|^2, \tag{8}$$

where $\bigtriangledown_1$ denotes the gradient with respect to the first argument. Under these conditions, $\|\boldsymbol{\lambda}_{k+1} - \boldsymbol{\lambda}_k\| \to 0$ when $0 < \beta_k < \alpha/L$. The choice of constant $\alpha$ is also discussed in [21]. Note that even though $h$ is required to be concave, $f$ could be non-convex. The lower bound usually contains concave terms, e.g., in the entropy term. In the worst case when there are no concave terms, we can simply choose $h \equiv 0$.

## 5 Examples of KL Proximal-Gradient Variational Inference

In this section, we show a few examples where the subproblem (7) has a closed-form solution.

**Generalized linear model :** We consider the generalized linear model shown in (9). Here, $\mathbf{y}$ is the output vector (of length $N$) whose $n$'th entry is equal to $y_n$, whereas $\mathbf{X}$ is an $N \times D$ feature matrix that contains feature vectors $\mathbf{x}_n^T$ as rows. The weight vector $\mathbf{z}$ is a Gaussian with mean $\boldsymbol{\mu}$ and covariance $\boldsymbol{\Sigma}$. To obtain the probability of $y_n$, the linear predictor $\mathbf{x}_n^T \mathbf{z}$ is passed through $p(y_n|\cdot)$.

$$p(\mathbf{y}, \mathbf{z}) := \prod_{n=1}^{N} p(y_n|\mathbf{x}_n^T \mathbf{z}) \mathcal{N}(\mathbf{z}|\boldsymbol{\mu}, \boldsymbol{\Sigma}). \tag{9}$$

We restrict the posterior distribution to be a Gaussian $q(\mathbf{z}|\boldsymbol{\lambda}) = \mathcal{N}(\mathbf{z}|\mathbf{m}, \mathbf{V})$ with mean $\mathbf{m}$ and covariance $\mathbf{V}$, therefore $\boldsymbol{\lambda} := \{\mathbf{m}, \mathbf{V}\}$. For this posterior family, the non-Gaussian terms $p(y_n|\mathbf{x}_n^T \mathbf{z})$ are difficult to handle, while the Gaussian term $\mathcal{N}(\mathbf{z}|\boldsymbol{\mu}, \boldsymbol{\Sigma})$ is easy because it is conjugate to $q$. Therefore, we set $\tilde{p}_e(\mathbf{z}|\boldsymbol{\lambda}) \equiv \mathcal{N}(\mathbf{z}|\boldsymbol{\mu}, \boldsymbol{\Sigma})/\mathcal{N}(\mathbf{z}|\mathbf{m}, \mathbf{V})$ and let the rest of the terms go in $\tilde{p}_d$.

By substituting in (6) and using the definition of the KL divergence, we get the lower bound shown below in (10). The first term is the function $f$ that will be linearized, and the second term is the function $h$.

$$\mathcal{L}(\mathbf{m}, \mathbf{V}) := \underbrace{\sum_{n=1}^{N} \mathbb{E}_{q(\mathbf{z}|\boldsymbol{\lambda})}[\log p(y_n|\mathbf{x}_n^T \mathbf{z})]}_{f(\boldsymbol{m}, \boldsymbol{V})} + \underbrace{\mathbb{E}_{q(\mathbf{z}|\boldsymbol{\lambda})} \left[ \log \frac{\mathcal{N}(\mathbf{z}|\boldsymbol{\mu}, \boldsymbol{\Sigma})}{\mathcal{N}(\mathbf{z}|\mathbf{m}, \mathbf{V})} \right]}_{h(\boldsymbol{m}, \boldsymbol{V})}. \tag{10}$$

For linearization, we compute the gradient of $f$ using the chain rule. Denote $f_n(\widetilde{m}_n, \widetilde{v}_n) := \mathbb{E}_{q(\mathbf{z}|\boldsymbol{\lambda})}[\log p(y_n|\mathbf{x}_n^T \mathbf{z})]$ where $\widetilde{m}_n := \mathbf{x}_n^T \mathbf{m}$ and $\widetilde{v}_n := \mathbf{x}_n^T \mathbf{V} \mathbf{x}_n$. Gradients of $f$ w.r.t. $\mathbf{m}$ and $\mathbf{V}$ can then be expressed in terms of gradients of $f_n$ w.r.t. $\widetilde{m}_n$ and $\widetilde{v}_n$:

$$\bigtriangledown_{\mathbf{m}} f(\mathbf{m}, \mathbf{V}) = \sum_{n=1}^{N} \mathbf{x}_n \bigtriangledown_{\widetilde{m}_n} f_n(\widetilde{m}_n, \widetilde{v}_n), \quad \bigtriangledown_{\mathbf{V}} f(\mathbf{m}, \mathbf{V}) = \sum_{n=1}^{N} \mathbf{x}_n \mathbf{x}_n^T \bigtriangledown_{\widetilde{v}_n} f_n(\widetilde{m}_n, \widetilde{v}_n), \tag{11}$$

For notational simplicity, we denote the gradient of $f_n$ at $\widetilde{m}_{nk} := \mathbf{x}_n^T \mathbf{m}_k$ and $\widetilde{v}_{nk} := \mathbf{x}_n^T \mathbf{V}_k \mathbf{x}_n$ by,

$$\alpha_{nk} := -\bigtriangledown_{\widetilde{m}_n} f_n(\widetilde{m}_{nk}, \widetilde{v}_{nk}), \quad \gamma_{nk} := -2 \bigtriangledown_{\widetilde{v}_n} f_n(\widetilde{m}_{nk}, \widetilde{v}_{nk}). \tag{12}$$

Using (11) and (12), we get the following linear approximation of $f$:

$$f(\mathbf{m}, \mathbf{V}) \approx \boldsymbol{\lambda}^T \bigtriangledown f(\boldsymbol{\lambda}_k) := \mathbf{m}^T [\bigtriangledown_{\mathbf{m}} f(\mathbf{m}_k, \mathbf{V}_k)] + \text{Tr}[\mathbf{V} \{\bigtriangledown_{\mathbf{V}} f(\mathbf{m}_k, \mathbf{V}_k)\}] \tag{13}$$

$$= -\sum_{n=1}^{N} \left[ \alpha_{nk} (\mathbf{x}_n^T \mathbf{m}) + \tfrac{1}{2} \gamma_{nk} (\mathbf{x}_n^T \mathbf{V} \mathbf{x}_n) \right]. \tag{14}$$

Substituting the above in (7), we get the following subproblem in the $k$'th iteration:

$$(\mathbf{m}_{k+1}, \mathbf{V}_{k+1}) = \arg \max_{\mathbf{m}, \mathbf{V} \succ 0} -\sum_{n=1}^{N} \left[ \alpha_{nk} (\mathbf{x}_n^T \mathbf{m}) + \tfrac{1}{2} \gamma_{nk} (\mathbf{x}_n^T \mathbf{V} \mathbf{x}_n) \right] + \mathbb{E}_{q(\mathbf{z}|\boldsymbol{\lambda})} \left[ \frac{\mathcal{N}(\mathbf{z}|\boldsymbol{\mu}, \boldsymbol{\Sigma})}{\mathcal{N}(\mathbf{z}|\mathbf{m}, \mathbf{V})} \right]$$

$$- \frac{1}{\beta_k} \mathbb{D}_{KL} \left[ \mathcal{N}(\mathbf{z}|\mathbf{m}, \mathbf{V}) \| \mathcal{N}(\mathbf{z}|\mathbf{m}_k, \mathbf{V}_k) \right], \tag{15}$$

Taking the gradient w.r.t. $\mathbf{m}$ and $\mathbf{V}$ and setting it to zero, we get the following closed-form solutions (details are given in the supplementary material):

$$\mathbf{V}_{k+1}^{-1} = r_k \mathbf{V}_k^{-1} + (1 - r_k) \left[ \boldsymbol{\Sigma}^{-1} + \mathbf{X}^T \mathrm{diag}(\boldsymbol{\gamma}_k) \mathbf{X} \right], \tag{16}$$

$$\mathbf{m}_{k+1} = \left[ (1 - r_k) \boldsymbol{\Sigma}^{-1} + r_k \mathbf{V}_k^{-1} \right]^{-1} \left[ (1 - r_k)(\boldsymbol{\Sigma}^{-1} \boldsymbol{\mu} - \mathbf{X}^T \boldsymbol{\alpha}_k) + r_k \mathbf{V}_k^{-1} \mathbf{m}_k \right], \tag{17}$$

where $r_k := 1/(1 + \beta_k)$ and $\boldsymbol{\alpha}_k$ and $\boldsymbol{\gamma}_k$ are vectors of $\alpha_{nk}$ and $\gamma_{nk}$ respectively, for all $k$.

**Computationally efficient updates :** Even though the updates are available in closed form, they are not efficient when dimensionality $D$ is large. In such a case, an explicit computation of $\mathbf{V}$ is costly because the resulting $D \times D$ matrix is extremely large. We now derive efficient updates that avoids an explicit computation of $\mathbf{V}$.

Our derivation involves two key steps. The first step is to show that $\mathbf{V}_{k+1}$ can be parameterized by $\boldsymbol{\gamma}_k$. Specifically, if we initialize $\mathbf{V}_0 = \boldsymbol{\Sigma}$, then we can show that:

$$\mathbf{V}_{k+1} = \left[ \boldsymbol{\Sigma}^{-1} + \mathbf{X}^T \mathrm{diag}(\widetilde{\boldsymbol{\gamma}}_{k+1}) \mathbf{X} \right]^{-1}, \text{ where } \widetilde{\boldsymbol{\gamma}}_{k+1} = r_k \widetilde{\boldsymbol{\gamma}}_k + (1 - r_k) \boldsymbol{\gamma}_k. \tag{18}$$

with $\widetilde{\boldsymbol{\gamma}}_0 = \boldsymbol{\gamma}_0$. A detailed derivation is given in the supplementary material.

The second key step is to express the updates in terms of $\widetilde{m}_n$ and $\widetilde{v}_n$. For this purpose, we define some new quantities. Let $\widetilde{\mathbf{m}}$ be a vector whose $n$'th entry is $\widetilde{m}_n$. Similarly, let $\widetilde{\mathbf{v}}$ be the vector of $\widetilde{v}_n$ for all $n$. Denote the corresponding vectors in the $k$'th iteration by $\widetilde{\mathbf{m}}_k$ and $\widetilde{\mathbf{v}}_k$, respectively. Finally, define $\widetilde{\boldsymbol{\mu}} = \mathbf{X}\boldsymbol{\mu}$ and $\widetilde{\boldsymbol{\Sigma}} = \mathbf{X}\boldsymbol{\Sigma}\mathbf{X}^T$.

Now, by using the fact that $\widetilde{\mathbf{m}} = \mathbf{X}\mathbf{m}$ and $\widetilde{\mathbf{v}} = \mathrm{diag}(\mathbf{X}\mathbf{V}\mathbf{X}^T)$ and by applying the Woodbury matrix identity, we can express the updates in terms of $\widetilde{\mathbf{m}}$ and $\widetilde{\mathbf{v}}$, as shown below (a detailed derivation is given in the supplementary material):

$$\widetilde{\mathbf{m}}_{k+1} = \widetilde{\mathbf{m}}_k + (1 - r_k)(\mathbf{I} - \widetilde{\boldsymbol{\Sigma}}\mathbf{B}_k^{-1})(\widetilde{\boldsymbol{\mu}} - \widetilde{\mathbf{m}}_k - \widetilde{\boldsymbol{\Sigma}}\boldsymbol{\alpha}_k), \text{ where } \mathbf{B}_k := \widetilde{\boldsymbol{\Sigma}} + [\mathrm{diag}(r_k \widetilde{\boldsymbol{\gamma}}_k)]^{-1},$$

$$\widetilde{\mathbf{v}}_{k+1} = \mathrm{diag}(\widetilde{\boldsymbol{\Sigma}}) - \mathrm{diag}(\widetilde{\boldsymbol{\Sigma}}\mathbf{A}_k^{-1}\widetilde{\boldsymbol{\Sigma}}), \text{ where } \mathbf{A}_k := \widetilde{\boldsymbol{\Sigma}} + [\mathrm{diag}(\widetilde{\boldsymbol{\gamma}}_k)]^{-1}. \tag{19}$$

Note that these updates depend on $\widetilde{\boldsymbol{\mu}}, \widetilde{\boldsymbol{\Sigma}}, \boldsymbol{\alpha}_k$, and $\boldsymbol{\gamma}_k$ (whose size only depends on $N$ and is independent of $D$). Most importantly, these updates avoid an explicit computation of $\mathbf{V}$ and only require storing $\widetilde{\mathbf{m}}_k$ and $\widetilde{\mathbf{v}}_k$, both of which scale linearly with $N$.

Also note that the matrix $\mathbf{A}_k$ and $\mathbf{B}_k$ differ only slightly and we can reduce computation by using $\mathbf{A}_k$ in place of $\mathbf{B}_k$. In our experiments, this does not create any convergence issues.

To assess convergence, we can use the optimality condition. By taking the norm of the derivative of $\mathcal{L}$ at $\mathbf{m}_{k+1}$ and $\mathbf{V}_{k+1}$ and simplifying, we get the following criteria: $\|\widetilde{\boldsymbol{\mu}} - \widetilde{\mathbf{m}}_{k+1} - \widetilde{\boldsymbol{\Sigma}}\boldsymbol{\alpha}_{k+1}\|_2^2 + \mathrm{Tr}[\widetilde{\boldsymbol{\Sigma}} \{ \mathrm{diag}(\widetilde{\boldsymbol{\gamma}}_k - \boldsymbol{\gamma}_{k+1} - 1) \} \widetilde{\boldsymbol{\Sigma}}] \leq \epsilon$, for some $\epsilon > 0$ (derivation is in the supplementary material).

**Linear-Basis Function Model and Gaussian Process :** The algorithm presented above can be extended to linear-basis function models by using the *weight-space view* presented in [22]. Consider a non-linear basis function $\phi(\mathbf{x})$ that maps a $D$-dimensional feature vector into an $N$-dimensional feature space. The generalized linear model of (9) is extended to a linear basis function model by replacing $\mathbf{x}_n^T \mathbf{z}$ with the *latent function* $g(\mathbf{x}) := \phi(\mathbf{x})^T \mathbf{z}$. The Gaussian prior on $\mathbf{z}$ then translates to a kernel function $\kappa(\mathbf{x}, \mathbf{x}') := \phi(\mathbf{x})^T \boldsymbol{\Sigma} \phi(\mathbf{x})$ and a mean function $\widetilde{\mu}(\mathbf{x}) := \phi(\mathbf{x})^T \boldsymbol{\mu}$ in the latent function space. Given input vectors $\mathbf{x}_n$, we define the kernel matrix $\widetilde{\boldsymbol{\Sigma}}$ whose $(i, j)$'th entry is equal to $\kappa(\mathbf{x}_i, \mathbf{x}_j)$ and the mean vector $\widetilde{\boldsymbol{\mu}}$ whose $i$'th entry is $\widetilde{\mu}(\mathbf{x}_i)$.

Assuming a Gaussian posterior distribution over the latent function $g(\mathbf{x})$, we can compute its mean $\widetilde{m}(\mathbf{x})$ and variance $\widetilde{v}(\mathbf{x})$ using the proximal-gradient algorithm. We define $\widetilde{\mathbf{m}}$ to be the vector of

---

**Algorithm 1** Proximal-gradient algorithm for linear basis function models and Gaussian process

---

**Given:** Training data $(\mathbf{y}, \mathbf{X})$, test data $\mathbf{x}_*$, kernel mean $\widetilde{\boldsymbol{\mu}}$, covariance $\widetilde{\boldsymbol{\Sigma}}$, step-size sequence $r_k$, and threshold $\epsilon$.
**Initialize:** $\widetilde{\mathbf{m}}_0 \leftarrow \widetilde{\boldsymbol{\mu}}$, $\widetilde{\mathbf{v}}_0 \leftarrow \text{diag}(\widetilde{\boldsymbol{\Sigma}})$ and $\widetilde{\boldsymbol{\gamma}}_0 \leftarrow \delta_1 \mathbf{1}$.
**repeat**
    For all $n$ in parallel: $\alpha_{nk} \leftarrow \nabla_{\widetilde{m}_n} f_n(\widetilde{m}_{nk}, \widetilde{v}_{nk})$ and $\gamma_{nk} \leftarrow \nabla_{\widetilde{v}_n} f_n(\widetilde{m}_{nk}, \widetilde{v}_{nk})$.
    Update $\widetilde{\mathbf{m}}_k$ and $\widetilde{\mathbf{v}}_k$ using (19).
    $\widetilde{\boldsymbol{\gamma}}_{k+1} \leftarrow r_k \widetilde{\boldsymbol{\gamma}}_k + (1 - r_k)\boldsymbol{\gamma}_k$.
**until** $\|\widetilde{\boldsymbol{\mu}} - \widetilde{\mathbf{m}}_k - \widetilde{\boldsymbol{\Sigma}}\boldsymbol{\alpha}_k\| + \text{Tr}[\widetilde{\boldsymbol{\Sigma}} \, \text{diag}(\widetilde{\boldsymbol{\gamma}}_k - \boldsymbol{\gamma}_{k+1} - \mathbf{1})\widetilde{\boldsymbol{\Sigma}}] > \epsilon$.
Predict test inputs $\mathbf{x}_*$ using (20).

---

$\widetilde{m}(\mathbf{x}_n)$ for all $n$ and similarly $\widetilde{\mathbf{v}}$ to be the vector of all $\widetilde{v}(\mathbf{x}_n)$. Following the same derivation as the previous section, we can show that the updates of (19) give us the posterior mean $\widetilde{\mathbf{m}}$ and variance $\widetilde{\mathbf{v}}$. These updates are the *kernalized* version of (16) and (17).

For prediction, we only need the converged value of $\boldsymbol{\alpha}_k$ and $\boldsymbol{\gamma}_k$, denoted by $\boldsymbol{\alpha}^*$ and $\boldsymbol{\gamma}^*$, respectively. Given a new input $\mathbf{x}_*$, define $\kappa_{**} := \kappa(\mathbf{x}_*, \mathbf{x}_*)$ and $\boldsymbol{\kappa}_*$ to be a vector whose $n$'th entry is equal to $\kappa(\mathbf{x}_n, \mathbf{x}_*)$. The predictive mean and variance can be computed as shown below:

$$\widetilde{v}(\mathbf{x}_*) = \kappa_{**} - \boldsymbol{\kappa}_*^T [\widetilde{\boldsymbol{\Sigma}} + (\text{diag}(\widetilde{\boldsymbol{\gamma}}^*))^{-1}]^{-1} \boldsymbol{\kappa}_* \quad , \quad \widetilde{m}(\mathbf{x}_*) = \widetilde{\mu}_* - \boldsymbol{\kappa}_*^T \boldsymbol{\alpha}^* \tag{20}$$

A pseudo-code is given in Algorithm 1. Here, we initialize $\widetilde{\boldsymbol{\gamma}}$ to a small constant $\delta_1$, otherwise solving the first equation might be ill-conditioned.

These updates also work for the Gaussian process (GP) models with a kernel $k(\mathbf{x}, \mathbf{x}')$ and mean function $\widetilde{\mu}(\mathbf{x})$, and for many other latent Gaussian models such as matrix factorization models.

# 6 Experiments and Results

We now present some results on the real data. Our goal is to show that our approach gives comparable results to existing methods and is easy to implement. We also show that, in some cases, our method is significantly faster than the alternatives due to the kernel trick.

We show results on three models: Bayesian logistic regression, GP classification with logistic likelihood, and GP regression with Laplace likelihood. For these likelihoods, expectations can be computed (almost) exactly, for which we used the methods described in [23, 24]. We use a fixed step-size of $\beta_k = 0.25$ and 1 for logistic and Laplace likelihoods, respectively.

We consider three datasets for each model. A summary is given in Table 1. These datasets can be found at the data repository[1] of LIBSVM and UCI.

**Bayesian Logistic Regression:** Results for Bayesian logistic regression are shown in Table 2. We consider two datasets. For 'a1a', $N > D$, and, for 'Colon', $N < D$. We compare our 'proximal' method to three other existing methods: the 'MAP' method which finds the mode of the penalized log-likelihood, the 'Mean-Field' method where the distribution is factorized across dimensions, and the 'Cholesky' method of [25]. We implemented these methods using 'minFunc' software by Mark Schmidt[2]. We used L-BFGS for optimization. All algorithms are stopped when optimality condition is below $10^{-4}$. We set the Gaussian prior to $\boldsymbol{\Sigma} = \delta \mathbf{I}$ and $\boldsymbol{\mu} = 0$. To set the hyperparameter $\delta$, we use cross-validation for MAP, and maximum marginal-likelihood estimate for the rest of the methods. As we compare running times as well, we use a common range of hyperparameter values for all methods. These values are shown in Table 1.

For Bayesian methods, we report the negative of the marginal likelihood approximation ('Neg-Log-Lik'). This is (the negative of) the value of the lower bound at the maximum. We also report the log-loss computed as follows: $- \sum_n \log \hat{p}_n / N$ where $\hat{p}_n$ are the predictive probabilities of the test data and $N$ is the total number of test-pairs. A lower value is better and a value of 1 is equivalent to random coin-flipping. In addition, we report the total time taken for hyperparameter selection.

| Model | Dataset | $N$ | $D$ | %Train | #Splits | Hyperparameter range |
|---|---|---|---|---|---|---|
| LogReg | a1a | 32,561 | 123 | 5% | 1 | $\delta$ = logspace(-3,1,30) |
| | Colon | 62 | 2000 | 50% | 10 | $\delta$ = logspace(0,6,30) |
| GP class | Ionosphere | 351 | 34 | 50% | 10 | for all datasets |
| | Sonar | 208 | 60 | 50% | 10 | $\log(l)$ = linspace(-1,6,15) |
| | USPS-3vs5 | 1,540 | 256 | 50% | 5 | $\log(\sigma)$ = linspace(-1,6,15) |
| GP reg | Housing | 506 | 13 | 50% | 10 | $\log(l)$ = linspace(-1,6,15) |
| | Triazines | 186 | 60 | 50% | 10 | $\log(\sigma)$ = linspace(-1,6,15) |
| | Space_ga | 3,106 | 6 | 50% | 1 | $\log(b)$ = linspace(-5,1,2) |

Table 1: A list of models and datasets. %Train is the % of training data. The last column shows the hyperparameters values ('linspace' and 'logspace' refer to Matlab commands).

| Dataset | Methods | Neg-Log-Lik | Log Loss | Time |
|---|---|---|---|---|
| a1a | MAP | — | 0.499 | 27s |
| | Mean-Field | 792.8 | 0.505 | 21s |
| | Cholesky | 590.1 | 0.488 | 12m |
| | Proximal | 590.1 | 0.488 | 7m |
| Colon | MAP | — | 0.78 (0.01) | 7s (0.00) |
| | Mean-Field | 18.35 (0.11) | 0.78 (0.01) | 15m (0.04) |
| | Proximal | 15.82 (0.13) | 0.70 (0.01) | 18m (0.14) |

Table 2: A summary of the results obtained on Bayesian logistic regression. In all columns, a lower values implies better performance.

For MAP, this is the total cross-validation time, whereas for Bayesian methods it is the time taken to compute 'Neg-Log-Lik' for all hyperparameters values over the whole range.

We summarize these results in Table 2. For all columns, a lower value is better. We see that for 'a1a', fully Bayesian methods perform slightly better than MAP. More importantly, the Proximal method is faster than the Cholesky method but obtains the same error and marginal likelihood estimate. For the Proximal method, we use updates of (17) and (16) because $D \ll N$, but even in this scenario, the Cholesky method is slow due to expensive line-search for a large number of parameters.

For the 'Colon' dataset, we use the update (19) for the Proximal method. We do not compare to the Cholesky method because it is too slow for the large datasets. In Table 2, we see that, our implementation is as fast as the Mean-Field method but performs significantly better.

Overall, with the Proximal method, we achieve the same results as the Cholesky method but take less time. In some cases, we can also match the running time of the Mean-Field method. Note that the Mean-Field method does not give bad predictions and the minimum value of log-loss are comparable to our approach. However, as Neg-Log-Lik values for the Mean-Field method are inaccurate, it ends up choosing a bad hyperparameter value. This is expected as the Mean-Field method makes an extreme approximation. Therefore, cross-validation is more appropriate for the Mean-Field method.

**Gaussian process classification and regression:** We compare the Proximal method to expectation propagation (EP) and Laplace approximation. We use the GPML toolbox for this comparison. We used a squared-exponential kernel for the Gaussian process with two scale parameters $\sigma$ and $l$ (as defined in GPML toolbox). We do a grid search over these hyperparameters. The grid values are given in Table 1. We report the log-loss and running time for each method.

The left plot in Figure 1 shows the log-loss for GP classification on USPS_3vs5 dataset, where the Proximal method shows very similar behaviour to EP. These results are summarized in Table 3. We see that our method performs similar to EP, sometimes a bit better. The running times of EP and the Proximal method are also comparable. The advantage of our approach is that it is easier to implement compared to EP and it is also numerically robust. The predictive probabilities obtained with EP and the Proximal method for 'USPS_3vs5' dataset are shown in the right plot of Figure 1. The horizontal axis shows the test examples in an ascending order; the examples are sorted according to their predictive probabilities obtained with EP. The probabilities themselves are shown in the y-axis. A higher value implies a better performance, therefore the Proximal method gives

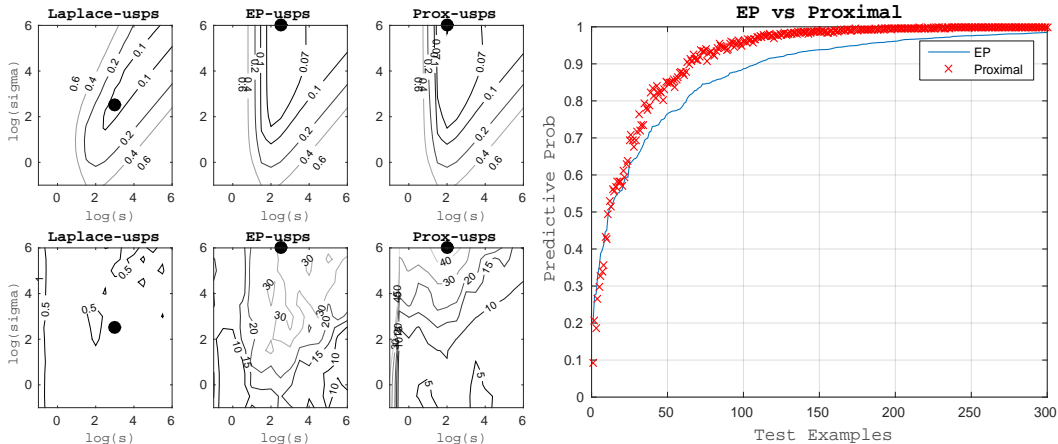

Figure 1: In the left figure, the top row shows the log-loss and the bottom row shows the running time in seconds for the 'USPS_3vs5' dataset. In each plot, the minimum value of the log-loss is shown with a black circle. The right figure shows the predictive probabilities obtained with EP and the Proximal method. The horizontal axis shows the test examples in an ascending order; the examples are sorted according to their predictive probabilities obtained with EP. The probabilities themselves are shown in the y-axis. A higher value implies a better performance, therefore the Proximal method gives estimates better than EP.

| | Log Loss | | | Time (s is sec, m is min, h is hr) | | |
|---|---|---|---|---|---|---|
| Data | Laplace | EP | Proximal | Laplace | EP | Proximal |
| Ionosphere | .285 (.002) | .234 (.002) | .230 (.002) | 10s (.3) | 3.8m (.10) | 3.6m (.10) |
| Sonar | .410 (.002) | .341 (.003) | .317 (.004) | 4s (.01) | 45s (.01) | 63s (.13) |
| USPS-3vs5 | .101 (.002) | .065 (.002) | .055 (.003) | 1m (.06) | 1h (.06) | 1h (.02) |
| Housing | 1.03 (.004) | .300 (.006) | .310 (.009) | .36m (.00) | 25m (.65) | 61m (1.8) |
| Triazines | 1.35 (.006) | 1.36 (.006) | 1.35 (.006) | 10s (.10) | 8m (.04) | 14m (.30) |
| Space_ga | 1.01 (—) | .767 (—) | .742 (—) | 2m (—) | 5h (—) | 11h (—) |

Table 3: Results for the GP classification using a logistic likelihood and the GP regression using a Laplace likelihood. For all rows, a lower value is better.

estimates better than EP. The improvement in the performance is due to the numerical error in the likelihood implementation. For the Proximal method, we use the method of [23], which is quite accurate. Designing such accurate likelihood approximations for EP is challenging.

# 7   Discussion and Future Work

In this paper, we have proposed a proximal framework that uses the KL proximal term to take the geometry of the posterior distribution into account. We established the equivalence between our proximal-point algorithm and natural-gradient methods. We proposed a proximal-gradient algorithm that exploits the structure of the bound to simplify the optimization. An important future direction is to apply stochastic approximations to approximate gradients. This extension is discussed in [21]. It is also important to design a line-search method to set the step sizes. In addition, our proximal framework can also be used for distributed optimization in variational inference [26, 11].

## Acknowledgments

Mohammad Emtiyaz Khan would like to thank Masashi Sugiyama and Akiko Takeda from University of Tokyo, Matthias Grossglauser and Vincent Etter from EPFL, and Hannes Nickisch from Philips Research (Hamburg) for useful discussions and feedback. Pierre Baqué was supported in part by the Swiss National Science Foundation, under the grant CRSII2-147693 "Tracking in the Wild".

## Footnotes

\*A note on contributions: P. Baqué proposed the use of the KL proximal term and showed that the resulting proximal steps have closed-form solutions. The rest of the work was carried out by M. E. Khan.

[1] https://archive.ics.uci.edu/ml/datasets.html and http://www.csie.ntu.edu.tw/~cjlin/libsvmtools/datasets/

[2] Available at https://www.cs.ubc.ca/~schmidtm/Software/minFunc.html

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
