[Supplementary Material]

# Supplementary Material for "Kullback-Leibler Proximal Variational Inference"

## 1 Proof of Theorem 1

The KL proximal point algorithm solves the following subproblems:

$$\boldsymbol{\lambda}_{k+1} = \arg\max_{\boldsymbol{\lambda}} \ \mathcal{L}(\boldsymbol{\lambda}) - \frac{1}{\beta_k} \mathbb{D}_{KL}[q(\mathbf{z}|\boldsymbol{\lambda}) \,\|\, q(\mathbf{z}|\boldsymbol{\lambda}_k)] \tag{1}$$

To prove the theorem, we will first derive the expression for the gradient descent updates using the natural gradient. Afterwards, we will derive the solution of (1) by differentiating the objective function. Afterwards, we do a few simplifications to obtain the theorem.

**Derivative of $\underline{\mathcal{L}}$:** Denote the mean-field update for $q_i$ by $\boldsymbol{\lambda}_i^*$. Then the gradient $\nabla_{\lambda_i}\mathcal{L}(\boldsymbol{\lambda})$ and the natural gradient $\widehat{\nabla}_{\lambda_i}\mathcal{L}(\boldsymbol{\lambda})$ are given as shown below (see Appendix A.1 and A.2 of [1] for a detailed derivation):

$$\nabla_{\lambda_i}\mathcal{L}(\boldsymbol{\lambda}) = \left[\nabla^2_{\lambda_i} A_i(\boldsymbol{\lambda}_i)\right](\boldsymbol{\lambda}_i^* - \boldsymbol{\lambda}_i) \quad , \quad \widehat{\nabla}_{\lambda_i}\mathcal{L}(\boldsymbol{\lambda}) = \boldsymbol{\lambda}_i^* - \boldsymbol{\lambda}_i. \tag{2}$$

Denoting the vector $\boldsymbol{\lambda}_i$ (or $\boldsymbol{\lambda}_i^*$) at $k$'th iteration by $\boldsymbol{\lambda}_{i,k}$ (or $\boldsymbol{\lambda}_{i,k}^*$), a gradient update along the natural gradient with step-size $\rho$ will result in the following update:

$$\boldsymbol{\lambda}_{i,k+1} \leftarrow \boldsymbol{\lambda}_{i,k} + \rho\widehat{\nabla}_{\lambda_i}\mathcal{L}(\boldsymbol{\lambda}) = (1-\rho)\boldsymbol{\lambda}_{i,k} + \rho\boldsymbol{\lambda}_{i,k}^* \tag{3}$$

**Solution of KL proximal-point algorithm:** We will now derive the solution of the proximal-point subproblem of (1). The gradient of the KL-divergence term can be derived using the definition of the KL-divergence for exponential family [2].

$$\mathbb{D}_{KL}[q_i(\mathbf{z}_i|\boldsymbol{\lambda}_i) \,\|\, q_i(\mathbf{z}_i|\boldsymbol{\lambda}_{i,k})] := A_i(\boldsymbol{\lambda}_{i,k}) - A_i(\boldsymbol{\lambda}_i) - \nabla_{\lambda_i}A_i(\boldsymbol{\lambda}_i)(\boldsymbol{\lambda}_{i,k} - \boldsymbol{\lambda}_i) \tag{4}$$

$$\Rightarrow \nabla_{\lambda_i}\mathbb{D}_{KL}[q_i(\mathbf{z}_i|\boldsymbol{\lambda}_i) \,\|\, q_i(\mathbf{z}_i|\boldsymbol{\lambda}_{i,k})] = -\nabla^2_{\lambda_i} A_i(\boldsymbol{\lambda}_i)(\boldsymbol{\lambda}_{i,k} - \boldsymbol{\lambda}_i) \tag{5}$$

The minimum of (1) can be obtained by setting the gradient to zero.

$$\nabla_{\lambda_i} \mathcal{L}(\boldsymbol{\lambda}) - \frac{1}{\beta_k} \nabla_{\lambda_i} \mathbb{D}_{KL}[q_i(\mathbf{z}_i|\boldsymbol{\lambda}_i) \,\|\, q_i(\mathbf{z}_i|\boldsymbol{\lambda}_{i,k})] = 0 \tag{6}$$

$$\Rightarrow \left[\nabla^2_{\lambda_i} A_i(\boldsymbol{\lambda}_i)\right](\boldsymbol{\lambda}_i^* - \boldsymbol{\lambda}_i) + \frac{1}{\beta_k} \nabla^2_{\lambda_i} A_i(\boldsymbol{\lambda}_i)(\boldsymbol{\lambda}_{i,k} - \boldsymbol{\lambda}_i) = 0 \tag{7}$$

$$\Rightarrow \left[\nabla^2_{\lambda_i} A_i(\boldsymbol{\lambda}_i)\right]\left[\boldsymbol{\lambda}_i^* - \boldsymbol{\lambda}_i + \frac{1}{\beta_k}(\boldsymbol{\lambda}_{i,k} - \boldsymbol{\lambda}_i)\right] = 0 \tag{8}$$

$$\Rightarrow \boldsymbol{\lambda}_{i,k+1} = \frac{1}{1 + \beta_k}\boldsymbol{\lambda}_{i,k} + \frac{\beta_k}{1 + \beta_k}\boldsymbol{\lambda}_{i,k}^* \tag{9}$$

Therefore, we see that $\rho = \beta_k/(1 + \beta_k)$.

## 2 Derivation for Generalized Linear Model

We will show that we obtain the following closed-form solutions,

$$\mathbf{V}_{k+1}^{-1} = r_k \mathbf{V}_k^{-1} + (1 - r_k)\left[\boldsymbol{\Sigma}^{-1} + \mathbf{X}^T\text{diag}(\boldsymbol{\gamma}_k)\mathbf{X}\right], \tag{10}$$

$$\mathbf{m}_{k+1} = \left[(1 - r_k)\boldsymbol{\Sigma}^{-1} + r_k\mathbf{V}_k^{-1}\right]^{-1}\left[(1 - r_k)(\boldsymbol{\Sigma}^{-1}\boldsymbol{\mu} - \mathbf{X}^T\boldsymbol{\alpha}_k) + r_k\mathbf{V}_k^{-1}\mathbf{m}_k\right], \tag{11}$$

for the following proximal-gradient subproblem:

$$(\mathbf{m}_{k+1}, \mathbf{V}_{k+1}) = \arg\max_{\mathbf{m}, \mathbf{V} \succ 0} -\sum_{n=1}^{N} \left[\alpha_{nk} \left(\mathbf{x}_n^T \mathbf{m}\right) + \tfrac{1}{2}\gamma_{nk} \left(\mathbf{x}_n^T \mathbf{V} \mathbf{x}_n\right)\right] + \mathbb{E}_{q(\mathbf{z}|\boldsymbol{\lambda})} \left[\frac{\mathcal{N}(\mathbf{z}|\boldsymbol{\mu}, \boldsymbol{\Sigma})}{\mathcal{N}(\mathbf{z}|\mathbf{m}, \mathbf{V})}\right]$$

$$-\frac{1}{\beta_k} \mathbb{D}_{KL}\left[\mathcal{N}(\mathbf{z}|\mathbf{m}, \mathbf{V})||\mathcal{N}(\mathbf{z}|\mathbf{m}_k, \mathbf{V}_k)\right]. \tag{12}$$

## 2.1   Update for $\mathbf{V}_{k+1}$

The KL divergence for Gaussian distribution is given as follows:

$$\mathbb{D}_{KL}\left[\mathcal{N}(\mathbf{z}|\mathbf{m}_1, \mathbf{V}_1)||\mathcal{N}(\mathbf{z}|\mathbf{m}_2, \mathbf{V}_2)\right] =$$
$$-\tfrac{1}{2}\left[\log|\mathbf{V}_1 \mathbf{V}_2^{-1}| - \text{Tr}(\mathbf{V}_1 \mathbf{V}_2^{-1}) - (\mathbf{m}_1 - \mathbf{m}_2)^T \mathbf{V}_2^{-1}(\mathbf{m}_1 - \mathbf{m}_2) + D\right] \tag{13}$$

Using this and the fact that the second term in (12) is the negative of the KL divergence, we expand (12) to get the following,

$$\tfrac{1}{2}[\log|\mathbf{V}| - \text{Tr}(\mathbf{V}\boldsymbol{\Sigma}^{-1}) - (\mathbf{m} - \boldsymbol{\mu})^T \boldsymbol{\Sigma}^{-1}(\mathbf{m} - \boldsymbol{\mu}) + D]$$

$$+ \frac{1}{2\beta_k}[\log|\mathbf{V}| - \text{Tr}\{\mathbf{V}(\mathbf{V}_k)^{-1}\} - (\mathbf{m} - \mathbf{m}_k)^T \mathbf{V}_k^{-1}(\mathbf{m} - \mathbf{m}_k) + D]$$

$$-\sum_{n=1}^{N}\left[\alpha_{nk}(\mathbf{x}_n^T \mathbf{m}) + \tfrac{1}{2}\gamma_{nk}(\mathbf{x}_n^T \mathbf{V}\mathbf{x}_n)\right] \tag{14}$$

$$= \tfrac{1}{2}\left[\left(1 + \frac{1}{\beta_k}\right)\log|\mathbf{V}| - \text{Tr}\left\{\mathbf{V}\left(\boldsymbol{\Sigma}^{-1} + \frac{1}{\beta_k}\mathbf{V}_k^{-1}\right)\right\} - (\mathbf{m} - \boldsymbol{\mu})^T \boldsymbol{\Sigma}^{-1}(\mathbf{m} - \boldsymbol{\mu})\right.$$

$$\left. -\frac{1}{\beta_k}(\mathbf{m} - \mathbf{m}_k)^T \mathbf{V}_k^{-1}(\mathbf{m} - \mathbf{m}_k) + \left(1 + \frac{1}{\beta_k}\right)D\right] - \sum_{n=1}^{N}\left[\alpha_{nk}(\mathbf{x}_n^T \mathbf{m}) + \tfrac{1}{2}\gamma_{nk}(\mathbf{x}_n^T \mathbf{V}\mathbf{x}_n)\right] \tag{15}$$

Taking the derivative with respect to $\mathbf{V}$ at $\mathbf{V} = \mathbf{V}_{k+1}$ and setting it to zero, we get the following:

$$\Rightarrow \quad \left(1 + \frac{1}{\beta_k}\right)\mathbf{V}_{k+1}^{-1} - \left(\boldsymbol{\Sigma}^{-1} + \frac{1}{\beta_k}\mathbf{V}_k^{-1}\right) - \mathbf{X}^T \text{diag}(\boldsymbol{\gamma}_k)\mathbf{X} = 0 \tag{16}$$

$$\Rightarrow \quad \mathbf{V}_{k+1}^{-1} = \frac{1}{1 + \beta_k}\mathbf{V}_k^{-1} + \frac{\beta_k}{1 + \beta_k}\left[\boldsymbol{\Sigma}^{-1} + \mathbf{X}^T \text{diag}(\boldsymbol{\gamma}_k)\mathbf{X}\right] \tag{17}$$

$$\Rightarrow \quad \mathbf{V}_{k+1}^{-1} = r_k \mathbf{V}_k^{-1} + (1 - r_k)\left[\boldsymbol{\Sigma}^{-1} + \mathbf{X}^T \text{diag}(\boldsymbol{\gamma}_k)\mathbf{X}\right] \tag{18}$$

where $r_k := 1/(1 + \beta_k)$.

## 2.2   Update for $\mathbf{m}_{k+1}$

Taking derivative with respect to $\mathbf{m}$ at $\mathbf{m} = \mathbf{m}_{k+1}$ and setting it to zero, we get the following:

$$\Rightarrow \quad -\boldsymbol{\Sigma}^{-1}(\mathbf{m}_{k+1} - \boldsymbol{\mu}) - \frac{1}{\beta_k}\mathbf{V}_k^{-1}(\mathbf{m}_{k+1} - \mathbf{m}_k) - \mathbf{X}^T \boldsymbol{\alpha}_k = 0 \tag{19}$$

$$\Rightarrow \quad -\left[\boldsymbol{\Sigma}^{-1} + \frac{1}{\beta_k}\mathbf{V}_k^{-1}\right]\mathbf{m}_{k+1} + \left[\boldsymbol{\Sigma}^{-1}\boldsymbol{\mu} + \frac{1}{\beta_k}\mathbf{V}_k^{-1}\mathbf{m}_k\right] - \mathbf{X}^T \boldsymbol{\alpha}_k = 0 \tag{20}$$

$$\Rightarrow \quad \mathbf{m}_{k+1} = \left[\boldsymbol{\Sigma}^{-1} + \frac{1}{\beta_k}\mathbf{V}_k^{-1}\right]^{-1}\left[\boldsymbol{\Sigma}^{-1}\boldsymbol{\mu} + \frac{1}{\beta_k}\mathbf{V}_k^{-1}\mathbf{m}_k - \mathbf{X}^T \boldsymbol{\alpha}_k\right] \tag{21}$$

$$\Rightarrow \quad \mathbf{m}_{k+1} = \left[\boldsymbol{\Sigma}^{-1} + \frac{1}{\beta_k}\mathbf{V}_k^{-1}\right]^{-1}\left[\boldsymbol{\Sigma}^{-1}\boldsymbol{\mu} + \frac{1}{\beta_k}\mathbf{V}_k^{-1}\mathbf{m}_k - \mathbf{X}^T \boldsymbol{\alpha}_k\right] \tag{22}$$

$$\Rightarrow \quad \mathbf{m}_{k+1} = \left[(1 - r_k)\boldsymbol{\Sigma}^{-1} + r_k \mathbf{V}_k^{-1}\right]^{-1}\left[(1 - r_k)\left(\boldsymbol{\Sigma}^{-1}\boldsymbol{\mu} - \mathbf{X}^T \boldsymbol{\alpha}_k\right) + r_k \mathbf{V}_k^{-1}\mathbf{m}_k\right] \tag{23}$$

where the last step is obtained using the fact that $1/\beta_k = r_k/(1 - r_k)$.

# 3 Derivation of the Computationally Efficient Updates

## 3.1 The first key step: reparameterization of $\mathbf{V}_{k+1}$

We show that $\mathbf{V}_{k+1}$ can be expressed in terms of $\boldsymbol{\gamma}_k$. Specifically, if we assume that $\mathbf{V}_0 = \boldsymbol{\Sigma}$, then

$$\mathbf{V}_{k+1} = \left[\boldsymbol{\Sigma}^{-1} + \mathbf{X}^T \mathrm{diag}(\widetilde{\boldsymbol{\gamma}}_{k+1})\mathbf{X}\right]^{-1}, \text{ where } \widetilde{\boldsymbol{\gamma}}_{k+1} = r_k\widetilde{\boldsymbol{\gamma}}_k + (1 - r_k)\boldsymbol{\gamma}_k. \tag{24}$$

with $\widetilde{\boldsymbol{\gamma}}_0 = \boldsymbol{\gamma}_0$.

We recursively substitute $\mathbf{V}_j$ for $j < k+1$ and simplify to get a convenient update.

$$\mathbf{V}_{k+1}^{-1} = r_k\mathbf{V}_k^{-1} + (1 - r_k)\left[\boldsymbol{\Sigma}^{-1} + \mathbf{X}^T\mathrm{diag}(\boldsymbol{\gamma}_k)\mathbf{X}\right] \tag{25}$$

$$= r_k\left[r_{k-1}\mathbf{V}_{k-1}^{-1} + (1 - r_{k-1})\left(\boldsymbol{\Sigma}^{-1} + \mathbf{X}^T\mathrm{diag}(\boldsymbol{\gamma}_{k-1})\mathbf{X}\right)\right] + (1 - r_k)\left(\boldsymbol{\Sigma}^{-1} + \mathbf{X}^T\mathrm{diag}(\boldsymbol{\gamma}_k)\mathbf{X}\right) \tag{26}$$

$$= r_kr_{k-1}\mathbf{V}_{k-1}^{-1} + r_k(1 - r_{k-1})\left(\boldsymbol{\Sigma}^{-1} + \mathbf{X}^T\mathrm{diag}(\boldsymbol{\gamma}_{k-1})\mathbf{X}\right) + (1 - r_k)\left(\boldsymbol{\Sigma}^{-1} + \mathbf{X}^T\mathrm{diag}(\boldsymbol{\gamma}_k)\mathbf{X}\right) \tag{27}$$

$$= r_kr_{k-1}\mathbf{V}_{k-1}^{-1} + (1 - r_kr_{k-1})\boldsymbol{\Sigma}^{-1} + \mathbf{X}^T\left[r_k(1 - r_{k-1})\mathrm{diag}(\boldsymbol{\gamma}_{k-1}) + (1 - r_k)\mathrm{diag}(\boldsymbol{\gamma}_k)\right]\mathbf{X} \tag{28}$$

$$= r_kr_{k-1}\left[r_{k-2}\mathbf{V}_{k-2}^{-1} + (1 - r_{k-2})\left(\boldsymbol{\Sigma}^{-1} + \mathbf{X}^T\mathrm{diag}(\boldsymbol{\gamma}_{k-2})\mathbf{X}\right)\right]$$
$$+ (1 - r_kr_{k-1})\boldsymbol{\Sigma}^{-1} + \mathbf{X}^T\left[r_k(1 - r_{k-1})\mathrm{diag}(\boldsymbol{\gamma}_{k-1}) + (1 - r_k)\mathrm{diag}(\boldsymbol{\gamma}_k)\right]\mathbf{X} \tag{29}$$

$$= r_kr_{k-1}r_{k-2}\mathbf{V}_{k-2}^{-1} + (r_kr_{k-1} - r_kr_{k-1}r_{k-2})\boldsymbol{\Sigma}^{-1} + (1 - r_kr_{k-1})\boldsymbol{\Sigma}^{-1}$$
$$+ \mathbf{X}^T\left[r_kr_{k-1}(1 - r_{k-2})\mathrm{diag}(\boldsymbol{\gamma}_{k-2}) + r_k(1 - r_{k-1})\mathrm{diag}(\boldsymbol{\gamma}_{k-1}) + (1 - r_k)\mathrm{diag}(\boldsymbol{\gamma}_k)\right]\mathbf{X} \tag{30}$$

$$= r_kr_{k-1}r_{k-2}\mathbf{V}_{k-2}^{-1} + (1 - r_kr_{k-1}r_{k-2})\boldsymbol{\Sigma}^{-1}$$
$$+ \mathbf{X}^T\left[r_kr_{k-1}(1 - r_{k-2})\mathrm{diag}(\boldsymbol{\gamma}_{k-2}) + r_k(1 - r_{k-1})\mathrm{diag}(\boldsymbol{\gamma}_{k-1}) + (1 - r_k)\mathrm{diag}(\boldsymbol{\gamma}_k)\right]\mathbf{X} \tag{31}$$

Continuing in this fashion until $k = 0$, we can write the update as follows:

$$\mathbf{V}_{k+1}^{-1} = t_k\mathbf{V}_0^{-1} + (1 - t_k)\boldsymbol{\Sigma}^{-1} + \mathbf{X}^T\mathrm{diag}(\boldsymbol{\gamma}_k)\mathbf{X} \tag{32}$$

where $t_k$ is the product of $r_k, r_{k-1}, \ldots, r_0$ and $\widetilde{\boldsymbol{\gamma}}_k$ is computed according to the following recursion:

$$\widetilde{\boldsymbol{\gamma}}_k = r_k\widetilde{\boldsymbol{\gamma}}_{k-1} + (1 - r_k)\boldsymbol{\gamma}_k \tag{33}$$

with $\widetilde{\boldsymbol{\gamma}}_{-1} = \boldsymbol{\gamma}_0$. If we set $\mathbf{V}_0 = \boldsymbol{\Sigma}$, then the formula simplifies to the following:

$$\mathbf{V}_{k+1}^{-1} = \boldsymbol{\Sigma}^{-1} + \mathbf{X}^T\mathrm{diag}(\widetilde{\boldsymbol{\gamma}}_k)\mathbf{X} \tag{34}$$

## 3.2 The second key step: expressing the updates in terms of $\widetilde{\mathbf{m}}$ and $\widetilde{\mathbf{v}}$

We recall the definition described in the paper. Define $\widetilde{\mathbf{m}}$ to be a vector with $\widetilde{m}_n$ as its $n$'th entry. Similarly, let $\widetilde{\mathbf{v}}$ be the vector of $\widetilde{v}_n$ for all $n$. Denote the corresponding vectors in the $k$'th iteration by $\widetilde{\mathbf{m}}_k$ and $\widetilde{\mathbf{v}}_k$, respectively. Let $\boldsymbol{\alpha}_k$ be the vector of $\alpha_{nk}$ for all $n$ and similarly let $\boldsymbol{\gamma}_k$ be the vector of $\gamma_{nk}$ for all $n$. Finally, define $\widetilde{\boldsymbol{\mu}} = \mathbf{X}\boldsymbol{\mu}$ and $\widetilde{\boldsymbol{\Sigma}} = \mathbf{X}\boldsymbol{\Sigma}\mathbf{X}^T$.

We will derive the following computationally efficient updates:

$$\widetilde{\mathbf{m}}_{k+1} = \widetilde{\mathbf{m}}_k + (1 - r_k)(\mathbf{I} - \widetilde{\boldsymbol{\Sigma}}\mathbf{B}_k^{-1})(\widetilde{\boldsymbol{\mu}} - \widetilde{\mathbf{m}}_k - \widetilde{\boldsymbol{\Sigma}}\boldsymbol{\alpha}_k), \text{ where } \mathbf{B}_k := \widetilde{\boldsymbol{\Sigma}} + [\mathrm{diag}(r_k\widetilde{\boldsymbol{\gamma}}_k)]^{-1},$$
$$\widetilde{\mathbf{v}}_{k+1} = \mathrm{diag}(\widetilde{\boldsymbol{\Sigma}}) - \mathrm{diag}(\widetilde{\boldsymbol{\Sigma}}\mathbf{A}_k^{-1}\widetilde{\boldsymbol{\Sigma}}), \text{ where } \mathbf{A}_k := \widetilde{\boldsymbol{\Sigma}} + [\mathrm{diag}(\widetilde{\boldsymbol{\gamma}}_k)]^{-1}. \tag{35}$$

We use the fact that $\widetilde{\mathbf{v}} = \text{diag}(\mathbf{X}\mathbf{V}\mathbf{X}^T)$ and apply Woodbury matrix identity.

$$\widetilde{\mathbf{v}}_{k+1} = \text{diag}(\mathbf{X}\mathbf{V}_{k+1}\mathbf{X}^T) = \text{diag}\left[\mathbf{X}(\mathbf{\Sigma}^{-1} + \mathbf{X}^T\text{diag}(\widetilde{\boldsymbol{\gamma}}_k)\mathbf{X})^{-1}\mathbf{X}^T\right] \tag{36}$$

$$= \text{diag}\left[\mathbf{X}\left\{\mathbf{\Sigma} - \mathbf{\Sigma}\mathbf{X}^T\left(\text{diag}(\widetilde{\boldsymbol{\gamma}}_k)^{-1} + \mathbf{X}\mathbf{\Sigma}\mathbf{X}^T\right)^{-1}\mathbf{X}\mathbf{\Sigma}\right\}\mathbf{X}^T\right] \tag{37}$$

$$= \text{diag}\left[\widetilde{\mathbf{\Sigma}} - \widetilde{\mathbf{\Sigma}}\left(\text{diag}(\widetilde{\boldsymbol{\gamma}}_k)^{-1} + \widetilde{\mathbf{\Sigma}}\right)^{-1}\widetilde{\mathbf{\Sigma}}\right] \tag{38}$$

$$= \text{diag}(\widetilde{\mathbf{\Sigma}}) - \text{diag}(\widetilde{\mathbf{\Sigma}}\mathbf{A}_k^{-1}\widetilde{\mathbf{\Sigma}}), \text{ where } \mathbf{A}_k := \widetilde{\mathbf{\Sigma}} + [\text{diag}(\widetilde{\boldsymbol{\gamma}}_k)]^{-1}. \tag{39}$$

Now we derive updates for $\widetilde{\mathbf{m}}_{k+1}$. First, we simply the updates of $\mathbf{m}_{k+1}$ as shown below. The first step is obtained by adding and subtracting $(1 - r_k)\mathbf{\Sigma}^{-1}\mathbf{m}_k$ in the square bracket at the right. In the second step, we take out $\mathbf{m}_k$. The final step is obtained by plugging in the updates of $\mathbf{V}_k$.

$$\mathbf{m}_{k+1} = \left[(1 - r_k)\mathbf{\Sigma}^{-1} + r_k\mathbf{V}_k^{-1}\right]^{-1}\left[(1 - r_k)(\mathbf{\Sigma}^{-1}\boldsymbol{\mu} - \mathbf{X}^T\boldsymbol{\alpha}_k) + r_k\mathbf{V}_k^{-1}\mathbf{m}_k\right] \tag{40}$$

$$= \left[(1 - r_k)\mathbf{\Sigma}^{-1} + r_k\mathbf{V}_k^{-1}\right]^{-1}\left[(1 - r_k)\{\mathbf{\Sigma}^{-1}(\boldsymbol{\mu} - \mathbf{m}_k) - \mathbf{X}^T\boldsymbol{\alpha}_k\} + \{(1 - r_k)\mathbf{\Sigma}^{-1} + r_k\mathbf{V}_k^{-1}\}\mathbf{m}_k\right] \tag{41}$$

$$= \mathbf{m}_k + (1 - r_k)\left[(1 - r_k)\mathbf{\Sigma}^{-1} + r_k\mathbf{V}_k^{-1}\right]^{-1}\left[\mathbf{\Sigma}^{-1}(\boldsymbol{\mu} - \mathbf{m}_k) - \mathbf{X}^T\boldsymbol{\alpha}_k\right] \tag{42}$$

$$= \mathbf{m}_k + (1 - r_k)\left[\mathbf{\Sigma}^{-1} + r_k\mathbf{X}^T\text{diag}(\widetilde{\boldsymbol{\gamma}}_{k-1})\mathbf{X}\right]^{-1}\left[\mathbf{\Sigma}^{-1}(\boldsymbol{\mu} - \mathbf{m}_k) - \mathbf{X}^T\boldsymbol{\alpha}_k\right] \tag{43}$$

Now we multiply the whole equation by $\mathbf{X}$ and use the fact that $\widetilde{\mathbf{m}} = \mathbf{X}\mathbf{m}$.

$$\widetilde{\mathbf{m}}_{k+1} = \widetilde{\mathbf{m}}_k + (1 - r^k)\mathbf{X}\left[\mathbf{\Sigma}^{-1} + r_k\mathbf{X}^T\text{diag}(\widetilde{\boldsymbol{\gamma}}_{k-1})\mathbf{X}\right]^{-1}\left[\mathbf{\Sigma}^{-1}(\boldsymbol{\mu} - \mathbf{m}_k) - \mathbf{X}^T\boldsymbol{\alpha}_k\right] \tag{44}$$

$$= \widetilde{\mathbf{m}}_k + (1 - r^k)\mathbf{X}\left\{\mathbf{\Sigma} - \mathbf{\Sigma}\mathbf{X}^T\left(\text{diag}(r_k\widetilde{\boldsymbol{\gamma}}_k)^{-1} + \mathbf{X}\mathbf{\Sigma}\mathbf{X}^T\right)^{-1}\mathbf{X}\mathbf{\Sigma}\right\}\left[\mathbf{\Sigma}^{-1}(\boldsymbol{\mu} - \mathbf{m}_k) - \mathbf{X}^T\boldsymbol{\alpha}_k\right] \tag{45}$$

$$= \widetilde{\mathbf{m}}_k + (1 - r^k)\left\{\mathbf{X}\mathbf{\Sigma} - \mathbf{X}\mathbf{\Sigma}\mathbf{X}^T\left(\text{diag}(r_k\widetilde{\boldsymbol{\gamma}}_k)^{-1} + \mathbf{X}\mathbf{\Sigma}\mathbf{X}^T\right)^{-1}\mathbf{X}\mathbf{\Sigma}\right\}\mathbf{\Sigma}^{-1}\left[\boldsymbol{\mu} - \mathbf{m}_k - \mathbf{\Sigma}\mathbf{X}^T\boldsymbol{\alpha}_k\right] \tag{46}$$

$$= \widetilde{\mathbf{m}}_k + (1 - r^k)\left\{\mathbf{X} - \widetilde{\mathbf{\Sigma}}\left(\text{diag}(r_k\widetilde{\boldsymbol{\gamma}}_k)^{-1} + \widetilde{\mathbf{\Sigma}}\right)^{-1}\mathbf{X}\right\}\left[\boldsymbol{\mu} - \mathbf{m}_k - \mathbf{\Sigma}\mathbf{X}^T\boldsymbol{\alpha}_k\right] \tag{47}$$

$$= \widetilde{\mathbf{m}}_k + (1 - r^k)\left\{\mathbf{I} - \widetilde{\mathbf{\Sigma}}\left(\text{diag}(r_k\widetilde{\boldsymbol{\gamma}}_k)^{-1} + \widetilde{\mathbf{\Sigma}}\right)^{-1}\right\}\left[\widetilde{\boldsymbol{\mu}} - \widetilde{\mathbf{m}}_k - \widetilde{\mathbf{\Sigma}}\boldsymbol{\alpha}_k\right] \tag{48}$$

$$= \widetilde{\mathbf{m}}_k + (1 - r_k)(\mathbf{I} - \widetilde{\mathbf{\Sigma}}\mathbf{B}_k^{-1})(\widetilde{\boldsymbol{\mu}} - \widetilde{\mathbf{m}}_k - \widetilde{\mathbf{\Sigma}}\boldsymbol{\alpha}_k) \tag{49}$$

where $\mathbf{B}_k := \widetilde{\mathbf{\Sigma}} + [\text{diag}(r_k\widetilde{\boldsymbol{\gamma}}_k)]^{-1}$.

## 4 Convergence Assessment

We will use the first-order condition which says that the gradient of $\mathcal{L}$ should be zero at the maximum. The lower bound is given as follows:

$$\mathcal{L}(\mathbf{m}, \mathbf{V}) = \sum_{n=1}^{N} f_n(\widetilde{m}_n, \widetilde{v}_n) + \mathbb{E}_{q(\mathbf{z}|\boldsymbol{\lambda})}\left[\frac{\mathcal{N}(\mathbf{z}|\boldsymbol{\mu}, \mathbf{\Sigma})}{\mathcal{N}(\mathbf{z}|\mathbf{m}, \mathbf{V})}\right] \tag{50}$$

$$= \sum_{n=1}^{N} f_n(\widetilde{m}_n, \widetilde{v}_n) + \frac{1}{2}[\log|\mathbf{V}| - \text{Tr}(\mathbf{V}\mathbf{\Sigma}^{-1}) - (\mathbf{m} - \boldsymbol{\mu})^T\mathbf{\Sigma}^{-1}(\mathbf{m} - \boldsymbol{\mu}) + D] \tag{51}$$

Taking the derivative w.r.t. $\mathbf{V}$ at $\mathbf{m} = \mathbf{m}_{k+1}, \mathbf{V} = \mathbf{V}_{k+1}$, we get the following:

$$\nabla_{\mathbf{V}}\mathcal{L}(\mathbf{m}, \mathbf{V}) = -\frac{1}{2}\mathbf{X}^T\text{diag}(\boldsymbol{\gamma}_{k+1})\mathbf{X} + \frac{1}{2}\mathbf{V}_{k+1}^{-1} - \frac{1}{2}\mathbf{\Sigma}^{-1} \tag{52}$$

$$= -\frac{1}{2}\mathbf{X}^T\text{diag}(\boldsymbol{\gamma}_{k+1})\mathbf{X} + \frac{1}{2}\left[\mathbf{\Sigma}^{-1} + \mathbf{X}^T\text{diag}(\widetilde{\boldsymbol{\gamma}}_k)\mathbf{X}\right] - \frac{1}{2}\mathbf{\Sigma}^{-1} \tag{53}$$

$$= \frac{1}{2}\mathbf{X}^T\left[\text{diag}(\widetilde{\boldsymbol{\gamma}}_k) - \text{diag}(\boldsymbol{\gamma}_{k+1})\right]\mathbf{X} - \frac{1}{2}\mathbf{\Sigma}^{-1}. \tag{54}$$

Taking the derivative w.r.t. $\mathbf{m}$ at $\mathbf{m} = \mathbf{m}_{k+1}, \mathbf{V} = \mathbf{V}_{k+1}$, we get:

$$\bigtriangledown_{\mathbf{m}}\mathcal{L}(\mathbf{m}, \mathbf{V}) = -\mathbf{X}^T\boldsymbol{\alpha}_{k+1} - \boldsymbol{\Sigma}^{-1}(\mathbf{m}_{k+1} - \boldsymbol{\mu}). \tag{55}$$

We can therefore monitor the two gradients to assess convergence:

$$\|\boldsymbol{\Sigma}^{-1}(\boldsymbol{\mu} - \mathbf{m}_{k+1}) - \mathbf{X}^T\boldsymbol{\alpha}_{k+1}\|_2^2 + \tfrac{1}{2}\mathrm{Tr}[\mathbf{X}^T\mathrm{diag}(\widetilde{\boldsymbol{\gamma}}_k - \boldsymbol{\gamma}_{k+1})\mathbf{X} - \boldsymbol{\Sigma}^{-1}] \quad \leq \epsilon, \tag{56}$$

To get computational efficient version, we can monitor the following:

$$\|\mathbf{X}\boldsymbol{\Sigma}\bigtriangledown_{\mathbf{m}}\mathcal{L}(\mathbf{m}, \mathbf{V})\|_2^2 + \mathrm{Tr}\Big[\mathbf{X}\boldsymbol{\Sigma}\bigtriangledown_{\mathbf{V}}\mathcal{L}(\mathbf{m}, \mathbf{V})\boldsymbol{\Sigma}\mathbf{X}^T\Big]$$
$$= |\widetilde{\boldsymbol{\Sigma}}\boldsymbol{\alpha}_{k+1} - \widetilde{\mathbf{m}}_{k+1} + \widetilde{\boldsymbol{\mu}}\|_2^2 + \mathrm{Tr}\Big[\widetilde{\boldsymbol{\Sigma}}\left\{\mathrm{diag}(\widetilde{\boldsymbol{\gamma}}_k - \boldsymbol{\gamma}_{k+1} - \mathbf{1})\right\}\widetilde{\boldsymbol{\Sigma}}\Big] \tag{57}$$