[Reviews · NeurIPS 2015]

Submitted by Assigned_Reviewer_1

The paper presents a general method for non-conjugate variational inference based on proximal method and linearisation of the non-conjugate model. This is shown to reduce to natural gradient optimisation for conjugate exponential models. The method is shown to lead to slightly better predictive accuracy than standard approximate inference methods in a few selected problems and data sets.

Quality

The method relies on linearisation to handle non-conjugate models. The seems potentially problematic, as previous works have found linearisation to be unreliable in variational inference with non-conjugate models (see e.g. Honkela and Valpola, NIPS 2004). The method is evaluated empirically on a few data sets, and mostly found to perform well. The paper focuses a lot on the positive aspects of the method, and its weaknesses and limitations are not mentioned at all.

Clarity

The paper is mostly clearly written and well-organised, but some key details are missing (specifically: what exactly is f_n).

Originality

The proposed approach is novel, although it is based on combination of existing techniques from various fields. Use of references is a bit shaky: the good fundamental reference from proximal algorithms seems to be missing and furthermore previous approaches using linearisation with non-conjugate variational inference are not discussed.

Significance

As the reliability of the proposed method is questionable, it is difficult to judge the paper's importance until this is resolved.

Other comments

1. The last sentence of the abstract contains unsubstantiated advertising: you method is not the best in every way. This must be clarified or deleted.

2. In Eqs. (2)-(3), what is eta?

3. In Eq. (3), why is it arg min and not arg max as in Eq. (2)?

4. As the proposed method seems to be a purely batch algorithm, it is unclear why it cites and links to SVI methods so much. Classic batch VB would seem much more relevant here.

5. You should clarify what f_n is precisely in the examples you study. Also, the accuracy of the linearisation procedure needs to be checked to make sure you do not suffer from similar problems as illustrated in Fig. 1 of Honkela and Valpola (NIPS 2004) for the Taylor series linearisation.

Additional references

Previous use of linearisation in variational inference:

Lappalainen, Harri, and Antti Honkela. "Bayesian non-linear independent component analysis by multi-layer perceptrons." In Advances in independent component analysis, pp. 93-121. Springer London, 2000.

Analysis of problems with linearisation:

Honkela, Antti, and Harri Valpola. "Unsupervised variational Bayesian learning of nonlinear models." In Advances in neural information processing systems, pp. 593-600. 2004.

Summary: A novel generic framework for non-conjugate variational inference. Presented results look OK, but some previous work raises doubt on the reliability of a key approximation.

Submitted by Assigned_Reviewer_2

This paper presented the use of proximal algorithms to perform variational inference.

The authors showed that natural gradient methods can be interpreted as a proximal point algorithm and also used proximal gradient methods (a.k.a mirror-descent) to non-conjugate settings.

The algorithm was demonstrated on non-conjugate models for regression and classification on real data sets.

Overall, this is a very nice paper that would be a great addition to the NIPS proceedings.

I think that the authors should definitely reference the recent work of Theis and Hoffman "A trust-regions method for stochastic variational inference with applications to streaming data" as their method is the proximal point method derived in this paper (though they do not explicitly say it's a proximal point method).

Additionally, there has been some recent work relating mirror-descent to Bayes theorem-like updates which Eqs. 11, 12, and 13 essential are (if \beta^k = 1), so some discussion of this would be nice as well.

Also, there's a somewhat abrupt transition from the proximal point viewpoint of variational inference to using proximal gradient methods and an explanation of this transition would be very helpful.

Lastly, there are a good amount of typos, both grammatical and mathematical, that definitely should be fixed with a careful proof-read.

Again, I think that this is a great paper that would fit in nicely at NIPS.
Summary: This is a nicely written paper that uses proximal algorithms to perform variational inference and is applicable to non-conjugate settings.

Other than some small criticisms I think that the paper would be a nice addition to the NIPS proceedings.

Submitted by Assigned_Reviewer_3

The submission introduces a modified proximal gradient algorithm for variational Bayes. The algorithm turns out to be equivalent to natural gradient descent with a certain choice of step sizes.

Comments on Quality:

The quality of the ideas in the paper is high, though not without flaws. The proximal gradient framework provides a clean way to deal with nonconjugacy and the equivalence to natural gradient descent serves as a justification and a useful link to a widely accepted method.

The flaws as far as ideas go are as follows. (1) As the authors note, the algorithm is actually not a proximal gradient algorithm because the KL term goes the wrong way. The motivation for doing things this way appears to be efficiency (the optimization problem decouples across parameters in the authors' setup, but not in the true proximal gradient setup). (2) Because of issue #1, the algorithm cannot be shown to converge except in special cases (e.g. beta_k --> 0 fast enough and in fact that seems necessary and sufficient).

Unfortunately, the submission does not sufficiently analyze the algorithm to demonstrate its utility. The primary argument for the algorithm appears to be potentially improved results in practice. Yet, the performance gains seem too small to justify the order of magnitude increase in runtime required to use the authors' method. Much stronger results, showing substantial improvements over the mean field baseline, would be necessary to make the submission compelling enough for acceptance.

On the whole, the reviewer believes the work has potential, but that it is too preliminary to be published in the conference.

Comments on Clarity:

Most of the paper is presented clearly, apart from Sections 5-6, which are nearly impossible to follow. The reviewer suggests the authors revise these to make them easier to understand.

Comments on Originality:

The work appears original.

Comments on Significance:

For now, the significance appears low. The algorithm could, however, have meaningful practical significance---further experiments are required to determine that.
Summary: The proximal gradient variational Bayes algorithm proposed in the submission is intriguing and warrants further investigation. The results presented in the submission, however, are not strong enough to justify accepting the paper.

Author Feedback
Author rebuttal: We thank all the reviewers. We realize that there are some confusions, perhaps due to our writing. We address some common issues below and then give more detailed response. We request reviewers to reconsider their reviews.

Regarding convergence: In this paper, our focus was primarily in showing usefulness of the method, therefore we did not present any convergence results (also space is limited). We do have a proof of convergence (for fixed step-sizes) for the algorithm of Section 5, which we will add in the appendix. Please see details in the response to Reviewer 3.

Linearisation: There is a confusion that our method is an approximation to the VB method. This is not correct. Linearization is the basis of prox-gradient methods, but it is employed only to take a gradient step. For the quadratic proximal distance, this is equivalent to gradient-descent, while in our case it is a natural-gradient step. Our method therefore converges to the maximum of the exact lower bound. Please see comments to reviewer 7.

Detailed comments:

REVIEWER 1 & 2: Thanks for the interesting references. We will also clarify explanations and add convergence proof to boost the theory part.

REVIEWER 3 & 4:

The comment about our algorithm 'not being a proximal method' is not correct. Perhaps, our comment in Sec 6 has led to the confusion. The method presented in Sec 5 is indeed a proximal method. For this, D(,) needs to be zero only when both arguments are equal. Paul Tseng in [7] uses this definition to prove convergence (see Thm 1 and Lemma 1 in Sec 4 of [7]). Another definition is given in this paper (see Def. 1).
- A New Perspective of Proximal Gradient Algorithms, Yi Zhoua, Yingbin Liangb, Lixin Shenc, ArXiv, 2015.

Our method is not a "conventional" prox-grad method since we do not use a Bregman divergence (i.e. mirror-descent). You correctly point out, that the
convergence of the method is not known (Tseng only proved prox-point, not prox-grad). We do have a proof of convergence for the method presented in Se
ction 5. We use results of Ghadimi et. al. [20]. The proof requires a small modification of their Lemma 1 resulting in a statement exactly equivalent
to their Thm 1 (and corollary 1). The result shows that the method converges under a "fixed step size".

We did not include the proof since it is long and not the main focus of the paper, but we will add it in the Appendix.

Clarification about (our second) primary argument: Please see Line 176-178. Our primary argument is that each step involves fitting a conjugate model with a closed-form expression, e.g. for GLM we only need linear-regression steps. Improvement might seem small, but this has big implications in practice e.g. we can now run large D and small N case without any mean-field approximation. Also, for GPs we don't need N^2 size matrices anymore etc.

Regarding comparison to Mean-Field: Comparisons with mean-field have been done previously e.g. Kuss and Rasmussen 2005 for binary GPs, therefore we do not include them for GPs. Also, the complexity of a full-covariance method is bound to be worse than mean-field, more so when the covariance matrix is large. Our contribution is that now we can run some large D cases without making mean-field approximation.

Clarity of Sec 5 and 6: Yes, this is a good point. We will shorten Sec 6 and provide more details in Sec 5.

REVIEWER 6: We will add Wang and Blei's method. Thanks.

REVIEWER 7:

Regarding f_n : We did not give details due to space constraints (and perhaps, a lack of definition for f_n is the source of the confusion about linearlisation). The definition is given in Line 193 and references mentioned in line 361, but we will add an explicit formula in the Appendix. For now, please see the Appendix E of reference [23]. We apologize for the trouble.

We do use Gauss-quadrature (implemented in GPML toolbox) to generalize computation of f_n, which is similar to your suggested reference.

Linearisation: Please see the comment at the top. Hopefully, this clarifies that there is no approximation during optimization.

Weakness of the method: Thanks for pointing out. We will add more details on this. A major limitation is the generalization to complex models since f_n is difficult to compute. We discuss this in future work, where we indicate that a stochastic-approx method will solve this problem.

Regarding References : We will add a discussion of linearization methods as per your suggestion. Thanks.

Sorry for the misleading statement in the abstract. This wasn't our intention. We will modify the sentence as per your suggestion.

About citing SVI: Our method is related to natural-gradients based methods including SVI and others (Honkela et. al., Sato et. al.), all cited in the paper. We will modify the text to make sure that our statements don't sound biased towards SVI only.